# Anti-Aggregative and Protective Effects of Vicenin-2 on Heat and Oxidative Stress-Induced Damage on Protein Structures

**DOI:** 10.3390/ijms242417222

**Published:** 2023-12-07

**Authors:** Giuseppe Tancredi Patanè, Lisa Lombardo, Stefano Putaggio, Ester Tellone, Silvana Ficarra, Davide Barreca, Giuseppina Laganà, Laura De Luca, Antonella Calderaro

**Affiliations:** Department of Chemical, Biological, Pharmaceutical and Environmental Science, University of Messina, 98166 Messina, Italy; giuseppe.patane@studenti.unime.it (G.T.P.); lisa.lombardo@studenti.unime.it (L.L.); stefano.putaggio@studenti.unime.it (S.P.); este.rtellone@unime.it (E.T.); silvana.ficarra@unime.it (S.F.); laura.deluca@unime.it (L.D.L.); anto.calderaro@gmail.com (A.C.)

**Keywords:** vicenin-2, anti-aggregative properties, protective effects against oxidative stress, spectroscopic analysis, molecular modeling

## Abstract

Vicenin-2, a flavonoid categorized as a flavones subclass, exhibits a distinctive and uncommon C-glycosidic linkage. Emerging evidence challenges the notion that deglycosylation is not a prerequisite for the absorption of C-glycosyl flavonoid in the small intestine. Capitalizing on this experimental insight and considering its biological attributes, we conducted different assays to test the anti-aggregative and antioxidant capabilities of vicenin-2 on human serum albumin under stressful conditions. Within the concentration range of 0.1–25.0 μM, vicenin-2 effectively thwarted the heat-induced HSA fibrillation and aggregation of HSA. Furthermore, in this study, we have observed that vicenin-2 demonstrated protective effects against superoxide anion and hydroxyl radicals, but it did not provide defense against active chlorine. To elucidate the underlying mechanisms, behind this biological activity, various spectroscopy techniques were employed. UV-visible spectroscopy revealed an interaction between HSA and vicenin-2. This interaction involves the cinnamoyl system found in vicenin-2, with a peak of absorbance observed at around 338 nm. Further evidence of the interaction comes from circular dichroism spectrum, which shows that the formation of bimolecular complex causes a reduction in α-helix structures. Fluorescence and displacement investigations indicated modifications near Trp214, identifying Sudlow’s site I, similarly to the primary binding site. Molecular modeling revealed that vicenin-2, in nonplanar conformation, generated hydrophobic interactions, Pi-pi stacking, and hydrogen bonds inside Sudlow’s site I. These findings expand our understanding of how flavonoids bind to HSA, demonstrating the potential of the complex to counteract fibrillation and oxidative stress.

## 1. Introduction

In the past, several studies have been published about the interaction between different molecules and proteins of the blood [1,2,3]. There is a growing interest in the formation of the bimolecular complex between the human serum albumin (HSA) and different exogenous and endogenous compounds. This interest stems from the many characteristics of HSA; in fact, it is a globular protein, and it is the most abundant protein present in the blood. Produced in the liver like pre-albumin, it is secreted by hepatocytes into the circulatory system, reaching a final concentration of 30/50 mg/mL with a half-life of 12–16 h. HSA has been extensively studied for its capacity to regulate oncotic pressure in order to reduce the oxidative processes and to bind and carry some different compounds [4]. This hypothetical link between HSA and various compounds may lead to changes in the structure of both components and new properties, such as the resistance to disease processes [5,6]. The focus of this article is to explore the potential interaction between HSA and a specific flavonoid, belonging to the subclass of flavones, called vicenin-2. It is also known as apigenin-6,8-di-C-glycopyranoside and isovitexin 8-C-glucoside and is characterized by an uncommon C-glycosidic linkage. Notably, increasing evidence suggests that deglycosylation is not a prerequisite for C-glycosyl flavonoid absorption in the small intestine. This results in the presence of the intact C-glycosyl flavonoids in human urine after oral consumption. In contrast to O-glycosyl flavonoids, the C-glycosidic linkage’s resistance to hydrolytic mechanisms in the upper- and mid-gastrointestinal tract, as well as during hepatic processing, leads to the excretion of intact C-glycosyl flavonoids in human urine [7,8]. The fundamental structure reveals the existence of an oxo group at the 4th position and a 2,3-double bond on ring C. This arrangement enables a conjugation between rings A and B that significantly influences the redox properties of these compounds. Furthermore, this central structure is adorned with the presence of two glucose molecules at the 6th and 8th positions of ring A. It represents a flavone *C*-glycoside that is a characteristic in the *Citrus* genus (found in fruits and juices), as well as the *Ocimum* genus, and certain medicinal plants with potential therapeutic properties such as *Urtica circularis*, *Perilla frutescens*, *Artemisia capillaries*, *Peperomia blanda*, *Potentilla discolor,* and various other flowers [9,10]. The positive effects of vicenin-2, encompassing antioxidant, antispasmodic, anti-hepatotoxic, and trypanocidal properties, as well as the enhancement of functional gastrointestinal and anti-nociceptive properties that contribute to a sense of well-being, have been documented in numerous studies [11,12,13,14]. Recently, vicenin-2, a major component of *U. circularis*, was found to hinder the production of inflammatory mediators such as tumor necrosis factor-α (TNF-α) and nitric oxide (NO) by inhibiting nuclear factor-kB (NF-kB), demonstrating the anti-inflammatory activity in the carrageen-induced rat paw edema model [15]. Additionally, vicenin-2 induced anti-angiogenic, pro-apoptotic, and anti-proliferative effects on prostate carcinoma cells. When combined with docetaxel, a preferred drug for treating androgen-independent prostate carcinoma, it synergistically hinders the growth of prostate tumors in rats in vivo to a greater extent than a single administration [16]. In this study, the interaction between vicenin-2 and HSA has been examined through a combination of multi-spectroscopic methods and docking studies. This investigation has yielded crucial insights, including the processes of a bimolecular complex. This complex induces a modification of the secondary protein structure of HSA, leading to a reduction in oxidative stress and the prevention of protein fibrillation. 

## 2. Results and Discussion

### 2.1. Protective Activity of Vicenin-2 against Protein Fibrillation and Oxidative Damage

Protein aggregation has gained much research attention in the last few decades due to implications of protein fibrillations for human health [17]. It is a process that is widespread and related to pathological but also normal proteins, which, under specific conditions, can undergo fibrillation with remarkable bioengineering applications. Proteins can undergo various processes, such as biological stresses, loss of native structure, or proteolysis, leading to the formation of self-aggregated structures. These structures, in turn, may contribute to the development of amyloid-fibrils, which are associated with several disorders [17,18]. Human serum albumin has been chosen to study the process because its physiological importance (it is one of the most abundant proteins and plays fundamental biologic roles; for example, it is the carrier protein of exogenous and endogenous compounds, as well as being a blood pressure regulator) and it has a well-defined and complex tertiary structure without any propensity to fibrillation in its native state; meanwhile, in vitro, it easily aggregates under different conditions, creating structures with features characteristic of disease-associated amyloidogenesis conditions. Different techniques have been employed to study processes such as the utilization of several crowding agents to mimic the packed interior or the solvent able to favor it. In our experimental conditions, we investigated the presence and morphology of HSA fibrils, after a 6-h incubation at 338 K, both in the absence or in the presence of different concentrations of vicenin-2 (0.0–25.0 µM). This examination utilized fluorescence microscopy and an interaction with Congo red. Following incubation at 338 K for 6 h, in the presence of 60% of ethanol (according to literature data [19]), well-defined fibrils of HSA were observed. These fibrils exhibited green fluorescence after thioflavine-T (ThT) staining, which was excited at a near-UV wavelength using a FITC filter in fluorescence microscopy, as depicted in Figure 1A. The experimental conditions induced protein-aggregation without varying pH and ionic strength values, resulting in the formation of HSA amyloid-like structures, which were characterized by a threadlike element with some branching points. The incubation in the presence of vicenin-2 induces changes in the process that are functions of the tested concentrations (Figure 1A). In the presence of 25.0 and 12.5 µM of the flavonoid, there was a complete inhibition of the fibrillation process, as also confirmed by UV-visible spectroscopy in the presence of Congo red; it had a spectrum that was completely superimposable to the ones obtained with HSA that was not incubated at a high temperature in the presence of the same concentration of vicenin-2. In particular, the Congo red assay was extensively employed to measure the fibril content of samples containing proteins, utilizing its spectral shift assay. From a chemical prospective, two binding sites of CR were identified in amyloid structures; one ran parallel to the β-sheet and the other ran antiparallel to the β-sheet [20]. Upon the interaction of the dey with the fibril structures, there was an increase in the absorption along with a bathochromic shift (red shift). In fact, the Congo red spectrum was characterized by two well-defined bands with a maximum between 340 and 486 nm (Figure 1B). Following the interaction with HSA fibrils, the maximum of the absorbance of the second band followed a red shift of about 24 nm (Figure 1B). The variation in the maximum of the absorbance with the HSA fibrils is depicted in Figure 1C. The statistical analysis confirmed the data obtained by fluorescence microscopy, and showed no statistically significant variation between the samples incubated at a high temperature in the presence of 25.0 and 12.5 µM of vicenin-2 and HSA incubated at 310 K. Low concentrations (6.2 and 3.1 µM) had no influence on the process with the appearance of HSA amyloid-like fibrils (Figure 1A), and had significant variations upon its interaction with Congo red (Figure 1A,C).

The potential formation of the HSA–vicenin-2 complex can also increase the stability of protein against oxidative stress. To analyze the process, we incubated the protein in the presence or absence of different concentrations of vicenin-2 (0–100 µM), and after the formation of the complex, we exposed the samples to superoxide anion and hydroxyl radicals, as well as active chlorine; these are known to be among the most prevalent radical species and oxidant species formed in circulation. After the electrophoretic run, gels were digitized with a camera, and the ImageJ program was used to quantify each band. Vicenin-2 is able to reduce the superoxide anion damage at concentrations ranging from 100.0 to 25.0 µM. The same result can be also observed against the hydroxyl radical (Figure 2). We observe a different behavior for the active chlorine, where vicenin-2 does not protect HSA from the oxidant, but, on the contrary, accentuates the damage to the protein. This is likely because vicenin-2 is not able to scavenge high- and low-molecular-mass and nitrogen-centered protein-derived radicals upon the reaction with active chlorine with lysine side-chain amino groups, causing the time-dependent fragmentation of the protein [21]. The activity against the superoxide anion and hydroxy radical are probably linked to the antioxidant activity of the molecule and its ability to scavenge these radicals, which is a function of the free radical nature and of the substituents present in the flavonoid (methoxy and hydroxyl groups, glycosidic moieties, and glycosylation) [22]. In fact, recently, Duan et al. [23] demonstrated that vicenin-2 has the capability to scavenge hydroxyl and DPPH radicals through the addition of hydrogen atoms.

### 2.2. Study of the Molecular Mechanism and Forces Involved in the Protective Activity of Vicenin-2

To shed light on the mechanism behind the protective activity of vicenin-2 against protein fibrillation and oxidative damage, we decided to analyze the interactions and the conformational changes involved in the process by UV-visible, fluorescence, molecular docking, and circular dichroism.

#### 2.2.1. UV-Visible Spectroscopy

The absorption spectrum of free vicenin-2 (76.5 µM) is marked by two primary absorption bands that are commonly denoted as Band I and Band II (Figure 3). Band I is linked to the absorption of the cinnamoyl system (B + C ring), while Band II is associated with the absorption of the benzoyl moiety formed by the conjugated system of ring A and ring C. The former is identified by an absorption band with a peak at 345 nm, and the latter has a maximum at 268 nm, along with a shoulder at 250 nm. Following the addition of increasing concentrations of HSA (0–76.5 µM), there is a light decrease in the maximum of absorbance band at 345 nm, with a concomitant increase in HSA (Figure 3). The inset of Figure 3 shows the average of the maximum absorbance of Band I, which was obtained in three different experiments. The titration of vicenin-2 with increasing HSA concentrations superior to 76.5 µM did not show any significant change in the absorption band, suggesting that all vicenin-2 molecules are already involved in the interaction with HSA. These results lead us to suppose that the interaction between the flavonoid and HSA can be described with 1:1 stoichiometry. The alterations in the absorption maximum of Band I entail the π-systems of both B- and C-rings, which are connected by a single C-C bond. The rotation around this C-C bond affects the intensity of the band and can be influenced by the conjugation between the two rings. In our investigation, the reduction in the intensity of the absorption maximum of Band I may stem from a decrease in the planarity of the two rings. This could occur as a consequence of the gradual inclusion of vicenin-2 into the HSA binding site, with the involvement of the C-ring and its phenol ring in the complex formation. This process could lead to the corresponding electronic rearrangement following the interaction with amino acid residues present in the binding site of the protein. The changes in the absorbance at 250 and 268 nm are negligible at all tested HSA concentrations. The spectra are also characterized by the presence of isosbestic points at 310 and 380 nm (Figure 3). 

#### 2.2.2. Fluorescence Spectroscopy

Fluorescence spectroscopy is one of the most utilized and sensitive tools, and is commonly utilized to analyze the changes in the fluorophore present in the macromolecules upon interaction with binding compounds or other proteins, suggesting useful information regarding the site of binding, residues involved in the formation of the complexes, thermodynamic forces, and binding mechanism. Among the 585 amino acids that composed the HSA primary structure, its fluorescence is due to the aromatic Trp 214 residue (located in the cavity of subdomain IIA of Sudlow’s site I) that is responsible for the maximum fluorescence of the protein, although the number of tyrosine (Tyr) and phenylalanine (Phe) residues is, by far, superior to the one of Trp. This is because phenylalanine shows a low quantum yield; its fluorescence is almost completely quenched by the presence of a nearby amino group or Trp residue [24,25], classifying HSA as a protein belonging to class B, as far as fluorescence is concerned. Fluorescence spectroscopy of proteins offers the opportunity to study many molecular interactions (such as a ground-state complex formation, energy transfer, molecular rearrangements, excited state reactions, or collisional quenching interactions). These interactions are elucidated through the quenching of protein fluorescence, which is typically categorized as dynamic quenching and static quenching. Dynamic quenching results from collisional processes, while static quenching is associated with a non-fluorescent ground-state complex formation [26].

In the present study, the interaction of vicenin-2 with HSA has been characterized, analyzing the quenching mechanism of protein fluorescence upon the interaction of the flavonoid at three different temperatures exciting HSA at 280 nm. Vicenin-2 did not show any fluorescence properties in the above-mentioned experimental conditions up to the maximum-tested concentrations (Figure 4). In Figure 4A, the steady-state fluorescence spectra of HSA alone (1.5 × 10^−5^ mol/L) or in the presence of the same molar concentration of vicenin-2 at 298 K is depicted. The emission maxima of HSA appeared at ~341 nm, while upon the addition of the same concentration of vicenin-2, the fluorescence intensity of the Trp residue of HSA showed a remarkable decrease, with a quenching of about 28%; the slight blue shift of the emission maxima of HSA (from 341 nm to 338 nm) suggested the formation of the complex. The increase in the hydrophobicity of the Trp microenvironment on the addition of the flavonoid was also supported by the hypochromic shift.

Based on this evidence, we applied the Stern−Volmer equation to elucidate the fluorescence quenching mechanism of vicenin-2:(1)$$\frac{F_{0}}{F}=1+K_{SV}\left[Q\right]=1+K_{q}\tau_{0}\left[Q\right]$$
where *F*_0_ and *F* represent the fluorescence intensities of HSA in the absence and presence of vicenin-2, respectively. *K_q_* is the quenching rate constant, *K_SV_* is the Stern−Volmer dynamic quenching constant, *τ*_0_ is the average lifetime of the fluorophore in the absence of quenchers, and [*Q*] is the concentration of the quencher. Stern-Volmer quenching constant (*K_SV_*) values at the tested temperatures were determined by analyzing the slope of the regression curve in the linear portion. These values are in line with those found for other flavonoids binding to HSA, as reported in literature [5,6,24,25,26,27]. In Figure 4B, the curves of F0/F versus [*Q*] at temperatures of 283, 304, and 310 K are reported. The K_SV_ values obtained at 283, 304, and 310 K are 2.75 (±0.10) × 10^4^, 2.32 (±0.09) × 10^4^ and 2.10 (±0.07) × 10^4^ M^−1^, respectively (Table 1). The K_SV_ values are inversely correlated with the temperature, indicating that the quenching of HSA fluorescence by this compound is due to a complex formation (static quenching), rather than by dynamic quenching [27]. Indeed, as documented in the literature, higher temperatures lead to faster diffusion and increased collisional quenching during the dynamic complex, resulting in a corresponding rise in the quenching constant; however, the opposite effect is anticipated for static quenching [27].

The fluorescence data was subsequently determined using a modified Stern−Volmer equation:(2)$$\frac{F_{0}}{F_{0}-F}=\frac{1}{f_{a}K_{a}\left[Q\right]}+\frac{1}{f_{a}}$$
where *K_a_* is the modified Stern−Volmer association constant for the accessible fluorophores, and fa is the fraction of accessible fluorescence. The linear regression analysis of *F*_0_/(*F*_0_ − *F*) versus 1/[*Q*] is depicted in Figure 4C. The obtained *K_a_* values are 1.95 × 10^4^, 1.58 × 10^4^, and 1.37 × 10^4^ M^−1^, at 298, 304, and 310 K, respectively (Table 1). The data analysis shows that the binding constant is not so high, and the effect of temperature is negligible. In fact, the degree of the quenching of flavonoids, upon binding to HSA, is related to the number as well as the position of the substituent and, in particular, the substituent groups at the ring-A and -C level (more substituents are present at ring-A with respect to ring-C, much higher is the quenching constant) and space steric hindrance [28]. Vicenin-2 shows the presence of four substituent groups in ring-A and one (-OH group) in ring-C, which may also probably be responsible for the negligible effects found in the function of the different temperatures analyzed. In fact, the formation of the hydrogen bond between amino acid residues of the protein and -OH group present in ring-C is one of the main responsibilities of the process and, in particular, major is the number of -OH group present in the ring-C much higher are the effects upon temperature changes. These data integrate the data of Diniz et al. [29] that reported a low percentage of binding to the HSA of vicenin-2 detected by capillary electrophoresis; they are not able to estimate the affinity constants and the numbers of binding primary sites because their obtained experimental data did not fit the equation utilized for the determination, due mainly to the low hydrophobicity of vicenin-2.

#### 2.2.3. Analysis of Binding Equilibrium, Thermodynamics, and Acting Forces 

The equilibrium between free and bound molecules, when small molecules bind to a set of equivalent sites on a macromolecule, can be described by the following equation:(3)$$log(F_{0}-F)/F=logK+nlog[Q]$$
where *K* is the observed binding constant to a site and n is the number of binding sites in each HSA macromolecules. The linear plot of log (*F*_0_ − *F*)/*F* as a function of *log*[*Q*] at 310 K is depicted in Figure 4D. The values of n are approximately equal to 1, suggesting a single class-binding site for vicenin-2; it is in the neighborhood of the tryptophan residue. The establishment of the protein–ligand complex is primarily defined by non-covalent interactions, including hydrophobic and electrostatic interactions, hydrogen bonds, and van der Waals forces. These interactions can be quantified through the analysis of thermodynamic parameters associated with the binding reaction. The van’t Hoff plot lets us calculate the thermodynamic parameters dependent on temperature and analyze the forces involved in the HSA–vicenin-2 complex formation. When the temperature variation is not so remarkable, the enthalpy change (Δ*H*) can be considered as a constant, and both the enthalpy change (Δ*H*) and entropy change (Δ*S*) can be determined from the van’t Hoff equation:(4)$$LnK_{LB}=-\frac{\Delta H}{RT}+\frac{\Delta S}{R}$$
where *R* is the gas constant. The enthalpy change (Δ*H*) is typically determined from the slope of the van’t Hoff plot (Figure 4E), while the entropy change (Δ*S*) is obtained from the intercept. The free energy change (Δ*G*) can be estimated using the equation:(5)ΔG=ΔH−TΔS

Figure 4E depicted the van’t Hoff plot for the evaluation of the thermodynamic parameters due to the HSA–vicenin-2 complex formation. The Δ*H* and Δ*S* values are 9.82 ± 0.93 kJ mol^−1^ and 5.6 ± 0.21 Jmol^−1^K^−1^, respectively. The Δ*G* at 298, 304, and 310 K is −11.49 ± 0.91, −11.52 ± 0.91, and −11.55 ± 0.90 kJmol^−1^, respectively. The positive variation of the ΔS value is due to the formation of hydrophobic interactions, while the negative value of ΔH suggested the involvement of hydrogen binding interactions and that the formation of the complex is predominately enthalpy driven [30]. The negative Δ*G* values, accompanied by the positive entropy change (Δ*S*), is indicative of a spontaneous process during the binding of vicenin-2 to HSA. 

#### 2.2.4. Fluorescence Displacement Binding Experiments

Displacement binding experiments have been carried out to analyze and identify the vicenin-2 binding site to HSA, using warfarin, ibuprofen, or digitoxin as the site marker, which univocally binds to the HSA binding site I, II, or III, respectively. The HSA–vicenin-2 complex was excited at the wavelength of 280 nm, and the changes in fluorescence were monitored, after the addition of the increasing concentration of each of the site markers at 341 nm. As can be observed in Figure 4F, the fluorescence of HSA–vicenin-2 was influenced by the addition of warfarin, while it remained almost the same in the presence of the other two site markers (ibuprofen and digitoxin). The competition between warfarin and vicenin-2 for the binding site to the protein (site I) supports the experimental evidence obtained by the above-reported experiments, in which vicenin-2 is most likely bound to the hydrophobic pocket located in subdomain IIA (the so called Sudlow’s site I). Recently, Rimac et al. have shown that, although flavonoids and warfarin show the same binding site, the region inside it in which they are binding is different [31]. This behavior is also present in our samples, with the maximum effects of displacement evident at a high concentration of warfarin (Figure 4F). The result, in which vicenin-2 and warfarin do not share the same binding region, is in line with the results published by other groups [32,33,34,35] that support the possibility of the IIA binding site to accommodate additional ligands in the warfarin proximity. 

#### 2.2.5. Molecular Docking Experiments

To confirm the experimental data collected and to gain information about the possible binding mode of vicenin-2, we have performed molecular docking experiments using the Induced Fit Docking (IFD) protocol [36] (Schrödinger Suite) to account for the flexibility of ligands and the receptor. In detail, we used the crystal structure of HSA in complex with the inhibitor warfarin (PDB ID: 1H9Z) [34]. Considering the experimental results, the molecule docking simulation box was set in the subdomain IIA (Sudlow’s site I).

To obtain results with high precision, the X-ray structure complex was submitted to a protocol validation. Firstly, we performed a rigid docking simulation through Glide [37]. The top-ranked pose, obtained from docking accurately, reproduced the binding mode in the crystal structure; in fact, the superimposition of the predicted pose to the crystallographic ligand showed a RMSD value of 0.83 Å (Figure 5A). Subsequently, we ran an IFD procedure on the best pose of the warfarin. Again, the docked solution is consistent with the crystallized ligand, showing an RMSD value of 1.50 Å. The increased RMSD is due to the torsion of the 2-propilketone moiety and a small rearrangement of the benzylic system; however, as showed in Figure 5B, the overall binding mode is preserved, demonstrating the reliability of the docking protocol. 

Considering the robustness of our docking protocol, we applied it to investigate the binding mode of vicenin-2. Again, the top-ranked pose produced by Glide was consequently subjected to the IFD calculation, and the Gscore best pose (−14.447 kcal/mol) was analyzed. 

The output data showed vicenin-2 placed in Sudlow’s site I; it is able to generate, with its cinnamoyl system, hydrophobic interactions with a subpocket formed by Leu198, Lys199, Ser202, Phe211, Trp214, Ala215, Arg218, and Leu481 (Figure 6B). A relevant interaction observed is the *pi-pi* stacking between the phenyl moiety of vicenin-2 and the indolic ring of Trp214. This finding aligns well with the results obtained by the fluorescence spectroscopy and UV-visible analysis, which emphasize the crucial role played by this interaction in maintaining the stability of the complex. The benzoyl system is surrounded by Arg218, Leu238, and His242, and engages direct contact with the side chain of Lys199 through pi-cation interactions. On the contrary, the beta-glucosyl units are exclusively involved in the instauration of the hydrogen bond.

To further explore the stability of the HSA–vicenin-2 complex, we submitted both the complex and free HSA to molecular dynamic (MD) simulations by using the Schrodinger Desmond module [38]. The Root Mean Square Deviation (RMSD) was calculated during the 20 ns simulation and revealed conformational changes in the protein (blue plot) until it stabilized at 13.5 ns (Figure 7A). Conversely, the RMSD of the apo protein shows a growing increase, proposing a low stability with respect to the HSA binding to vicenin-2 (Figure 7B). The RMSD progression ofvicenin-2 (magenta plot) is computed with respect to HSA and its binding site, revealing the stability of the ligand in the cavity during the time simulation (Figure 7A).

In Figure 8A, the histogram summarizes the total number of interactions between the residues and the ligand throughout the simulation. Each bar represents a value converted from a percentage rate to a decimal number. For instance, a value of 0.5 signifies a contact maintained for 50% of the simulation time. Additionally, values exceeding 1.0 are conceivable, given that certain residues may engage in multiple interactions of the same type with various atoms or groups of the ligand. Finally, Figure 8B highlights the specific subtypes of contacts that the residues establish with the atoms of the vicenin-2, considering the contacts that occur in more than 40%. The main contacts are formed with the same residues reported by the docking calculations. Interestingly, the most frequent interaction is established between the phenyl ring of the cinnamoyl system and the side chain of Trp214, supporting our information and indicating the essential role played in the binding of the ligand to the cavity. These results provide a plausible explanation for the binding mode of vicenin-2 to Sudlow’s site I and agree with the experimental evidence retrieved by the UV-visible and fluorescence spectroscopy.

### 2.3. Modification in the Conformation of HSA

The binding of small molecules to HSA can provide conformational changes that can be analyzed by CD spectroscopy due to changes in the intramolecular forces required to maintain the secondary structure. HSA is composed of 585 amino acids, and it has a molecular mass of 66,500 Da with 3 well defined and characterized homologous domains (domains I–III) divided into 2 subdomains (A and B), which are composed of 6 and 4 α-helices, respectively. As shown in UV-visible, fluorescence, and molecular docking experiments, the binding site of vicenin-2 to HSA is in Sudlow’s site I, and it is able to generate, with its cinnamoyl system, hydrophobic interactions with a subpocket composed by Leu198, Lys199, Ser202, Phe211, Trp214, Ala215, Arg218, and Leu481. Taking into account the formation of the weak interaction between vicenin-2 and HSA, CD spectroscopy has been employed to study the eventual secondary conformation changes upon the formation of the complex in the wavelength range of 200–260 nm at room temperature. Figure 9 shows the CD spectra of a solution of HSA at 2.8 µM, alone or in the presence of the same, double, and triple concentrations of vicenin-2.

The CD spectra of HSA is characterized by the presence of two well-defined negative bands at around 208 and 222 nm, which is typical of an α-helix secondary structure that is attributable to the contribution of both n → π* electronical transitions in amino acid residues and peptide bonds [39]. The interaction and formation of the complex with the flavonoid induce a slight change in the molar ellipticity of the protein, indicating little decrease in the content of *α*-helix structures of HSA. In particular, the loss of the *α*-helix upon a complex formation was estimated, according to Matei et al. [40], as follows:(6)$$\alpha -helix(\%)=\frac{{-MRE}_{222}-2340}{30300}\times 100$$
where MRE are the mean residue ellipticity, calculated as:(7)$$MRE=\frac{\theta }{10rl[HSA]}$$
where *θ* is the ellipticity observed by CD spectroscopy in millidegrees, *r* = 585 is the number of amino acid residues in has, and *l* is the path length of the cell (in cm). The analysis of the data shows the presence of about 55.6% α-helix secondary structures in the free HSA (according to data in the literature [41]), but upon the formation of the complex, there is a slight change in the HSA folding, with a loss of about 2.63 ± 0.11%, 3.31 ± 0.08, and 5.79 ± 0.12 of the α-helix utilizing the same, double and triple concentrations of vicenin-2 with respect to HSA. In the complex, the shape of the CD spectra did not show remarkable changes, indicating that the structure of HSA was still predominantly made up of α-helix structures, even when we utilized the double or triple concentration of the flavonoid. The conformational change suggested that the interaction between vicenin-2 and HSA induced a slight stretching of the polypeptide chain, changing the hydrogen bond network and the hydrophobic environment in Sudlow’s site, as also shown by the UV-visible and fluorescence data.

## 3. Materials and Methods

### 3.1. Reagents and Compounds

All reagents were supplied by Sigma–Aldrich (St. Louis, MO, USA) and used without further purification. Vicenin-2 was purchased by Merck (Darmstadt, Germany) and dissolved in dimethyl sulfoxide (DMSO).

### 3.2. HSA Fibrillation and Fluorescence Microscopy

To observe the possible protection of vicenin-2 in fibrillation experiments, we prepared a solution of HSA (0.05 mM) in 20 mM of sodium phosphate buffer with a pH of 7.4. To obtain a reference blank, a sample of HSA was incubated for 6 H at 210 K. To induce fibrillation, we incubated HSA samples for 6 h at 338 K in the absence or presence of an ethanol solution to increase the fibril formation, in the absence or presence of various concentrations of vicenin-2 (3.1, 6.25, 12.5, and 25.0 µM). We finished the incubation for 6 h at a high temperature, and the samples were analyzed after another 24 h of incubation at room temperature. A freshly prepared solution of ThT at a concentration of 1.0 mM was used for staining the samples. Specifically, 10 μL of the fluorescent solution was combined with 20 μL of each sample. This mixture was then placed on a glass slide covered with a coverslip. Thus, the stained samples were monitored using a Leica DM 2500 M microscope and a FITC filter ThT excitation and emission. The images were captured with a digital camera, and the observations were conducted at 20×/0.25.

### 3.3. Congo Red Assay

The investigation into the formation of aggregates, in the previously described samples, was investigated by UV-visible spectroscopy. This analysis was conducted by observing the shift in the absorbance of Congo red within the wavelength range from 400 to 700 nm. A Congo red stock solution (10 mM) was initially prepared by dissolving the dye in a solution of 10 mM sodium phosphate buffer (pH 7.0), containing 150 mM NaCl with continuous stirring. This stock solution was then filtered using 0.2 mm Millipore syringe filter. A fresh working solution was subsequently prepared by diluting the stock solution 100 times. For this study, 250 μL (27.2 μM) aliquots of the protein solutions were mixed with 250 μL of a solution containing 40 μM of Congo red. The total volume of the mixture (1 mL) was adjusted using 10 mM of a sodium phosphate buffer with a pH of 7.4.

### 3.4. Inhibition of Oxidative Stress against HSA 

To test the effects of vicenin-2 against oxidative stress-induced HSA damage, we exposed the protein to different radicals (anion superoxide, hydroxyl radical, and chloride radical) in the absence or presence of different concentrations of the flavonoid. In the experiment, we incubated HSA (15 μM) plus 0–100 μM of vicenin-2 in the absence or presence of an anion superoxide generated with the reaction of Nishiniky et al., using phenazine metasulfate (PMS) and NADH. In the second experiment, we used HSA (15 μM) plus 0–100 μM of vicenin-2 in the absence or presence of a hydroxyl radical generated with the Fenton reaction, using 0.025 mM of FeCl_3_, 0.104 mM of EDTA, 0.10 mM of ascorbic acid, and 2.8 mM of H_2_O_2_. In the last experiment, we used HSA (15 μM) plus 0–100 μM of vicenin-2 in the absence or presence of the chloride radical at a final concentration of 1.3% HClO. In all experiments, after the incubation of the samples for 1 h, we investigated the oxidative damage via 7.5% of the polyacrylamide-gel electrophoresis separation. After the electrophoretic run, all the gels were colorated with Coomassie Brilliant Blue; then, they were digitized and processed with the software ImageJ (available at the website http://rsb.info.nih.gov/ij/), which included background subtraction, contrast enhancement, dye front baseline correction, and signal-to-noise enhancement and presented as histograms 

### 3.5. UV-Visible Spectroscopy Spectra

UV-visible spectroscopy was performed on samples in buffer solutions (phosphate buffer, 20 mM, pH 7.4) to which an aliquot of the vicenin-2 stock solution (prepared in dimethyl sulfoxide) was added to obtain the desired final concentration. The study of the vicenin-2–HSA interaction was performed on a solution of vicenin-2 (with a final concentration of 76.5 µM), with an increasing amount of HSA up to 153.0 µM and while reading the changes in absorbance in the wavelength range of 240–440 nm, with a 1 cm optical path of quartz cuvette. The spectra of the buffer solution, alone or plus the same concentration of HSA that has been utilized for the titration, were subtracted from that of the corresponding sample solutions in the same experimental conditions. The experiment was repeated three-fold; the data was shown as mean ± SD, as far as the maximum of absorbance is concerned.

### 3.6. Fluorescence Spectra 

The fluorescence spectra were derived from analyses performed with a Horiba Jobin-Yvon FluoroMax-4 spectrofluorometer, which was equipped with a pulsed xenon lamp and fitted with a Peltier Thermoelectric Temperature Controller model LFI-3751 (5 A–40 W). Both excitation and emission bandwidths were fixed at 5 nm. Protein samples were excited at 280 and emission spectra were recorded in the range of 280–500 nm at 298, 304, and 310 K. To explore the potential interaction between HSA–vicenin-2, fluorometric titration was performed at different temperatures. In all experiments, the concentration of HSA was 1.5 × 10^−5^ mol/L HSA in 20 mM of sodium phosphate buffer (pH 7.4), and the concentration of vicenin-2 was varied from 0 to 9.0 × 10^−5^ mol/L HSA. All the fluorescence intensities were corrected for the inner filter effect utilizing the following formula:(8)$$F_{corr}=F_{obs}\times 10^{(Aex-Aem)/2}$$
where *F_corr_*, *F_obs_*, *Aex,* and *Aem* are the corrected and observed fluorescence intensity and the absorbance at an excitation and emission wavelength, respectively [42]. The experiment was repeated three-fold; the data were shown as mean ± SD as far as the maximum of absorbance is concerned.

### 3.7. Competitive Experiments

The binding site of vicenin-2, following the formation of the complex with HSA, was determined by performing competitive studies with several well-known markers such as warfarin, ibuprofen, and digoxin for sites I, II, and III, respectively. Increasing concentrations of site markers (1.0 × 10^−3^ mol/L) were added to a solution of equimolar concentrations of HSA and vicenin-2 (1.5 × 10^−5^ mol/L) in 20 mM of sodium phosphate buffer (pH 7.4) to obtain an overall site marker concentration value ranging from 0 to 9 × 10^−5^ mol/L. Fluorescence spectra were recorded at 310 K, as reported above. The fluorescence of the ternary mixture, following the excitation at 280 nm, as a percentage of the initial fluorescence, was determined according to the method of Sudlow et al. [43]: *F*_2_/*F*_1_ × 100(9)
where *F*_2_ and *F*_1_ denote the fluorescence intensities of the complex formed by HSA and vicenin-2, with or without the addition of the quencher, respectively, and at the wavelength of 341 nm.

### 3.8. Circular Dichroism (CD) Spectra Measurements

Circular dichroism spectra were obtained utilizing a Jasco J-810 CD spectrometer. The circular dichroism spectra of free HSA and the one of the HSA/vicenin-2 complex at the same molar ratio (2.8 × 10^−6^ M), or with the double (5.6 × 10^−6^ M) or triple (8.4 × 10^−6^ M) concentration of vicenin-2, and with respect to the one of HSA (2.8 × 10^−6^ M), were analyzed in the range of 195–300 nm in order to check the eventual changes in the secondary structure content of the protein. The measurements were performed twice at room temperature, corrected for the baseline by subtracting the spectral contribution of the buffer solution or the buffer plus vicenin-2, using a quartz cell of 1 mm. The obtained spectra were transformed utilizing the software peakfit seasolve v4.12, and the content of the secondary structure was obtained by analyzing the data according to Matei et al. [40]. The experiment was repeated three-fold, and the data were shown as mean ± SD as far as the content of the secondary structure is concerned.

### 3.9. Molecular Modeling Study

Docking analyses were carried out by means of the Schrödinger suite, LLC (Schrödinger Release 2020-4: Maestro, Schrödinger, LLC, New York, NY, USA. 2020) for the vicenin-2 and the co-crystalized warfarin. The 3D structure of vicenin-2 was taken from the ZINC database (entry code 4098604), and the warfarin was retrieved from its co-crystalized complex with HSA that is available on the Protein Data Bank (PDB ID: 1H9Z). Both ligands were submitted to a preparation protocol using the tool LigPrep of the Schrödinger suite (Schrödinger Release 2020-4: LigPrep, Schrödinger, LLC, New York, NY, USA. 2020). The preparation involved setting a pH value of 7.4 to mimic physiological conditions, utilize the OPLS3 force field, and maintain the original chiral center configuration. Similarly to the HSA model protein, we employed the structure co-crystalized with the warfarin (PDB ID: 1H9Z). The Protein Preparation Wizard module [44] was used to prepare the protein. This required removing water molecules and myristic acid units, adding hydrogens, assigning bond orders, and filling missing side chains. At this point, a minimization of the structure was performed to relieve any bad contact or strain. The rigid docking was calculated by means of Glide [37]. The receptor grid was generated via the Receptor Grid Generation tool, with the warfarin’s center identified as a centroid with a radius of 15 Å. Default parameters were applied for the Van der Waals radius scaling factor of 1.0 and partial charge cut-off of 0.25. The protocol used in the docking consists of the SP (Standard Precision) method as the fitness function, default Van der Waals scaling factor (scaling factor: 0.80; partial charge cut-off: 0.15), and OPLS3 force field. No constraints were defined. The number of reported poses for each ligand was limited to 10, and the best-ranked pose was selected for further steps. The Induced Fit Docking (IFD) [36] procedure includes Glide Docking, Prime Refinement [45], and Glide Redocking. The best rigid docking pose was used to define the centroid, and for the flexibility of side chains, carried out by Prime, the cut-off was established within 5 Å from the ligand. The parameters related to the scoring function and pose numbers to the output are the same set in the Ligand Docking method. The best score pose was retrieved for a further analysis. In the last step of our computational workflow, we performed a molecular dynamics (MD) simulation, both on the complex HSA–vicenin-2 and free HSA through the tool Desmond [38] implemented in the Schrödinger Suite. A size of 10 Å in x, y, and z directions was set for the simulation box, as well as an orthorhombic shape. The complex was soaked in the model solvent TIP3P, and the system was neutralized by adding NaCl ions at a concentration of 0.15 M. Again, OPLS3 was used as the force field. MD simulations were performed for 20 ns in the NPT ensemble at a temperature of 300 K with 1 atm of pressure, precisely.

### 3.10. Statistical Analysis

Data are presented as means ± standard deviation (S.D.). Data were analyzed by a one-way analysis of variance (ANOVA). The significance of the difference between the respective controls for each experimental test condition was assayed by using Tukey’s for each paired experiment. *p* < 0.05 was regarded as indicating a significant difference.

## 4. Conclusions

In the present study, we have demonstrated, for the first time, the anti-aggregative and protective effects of vicenin-2 on heat and oxidative stress-induced damage on protein structures in the order of a few µM concentrations (up to 25 µM). The analysis of the mechanism of action of the compound with a multi-spectroscopic approach and dynamic simulation highlights the involvement of moderate conformational changes of the protein, followed by the formation of the complex with a small decrease in the α-helix structure, as well as the involvement of both thermodynamic and weak interactions. Specifically, the analysis highlighted alterations in the environment surrounding Trp214, the primary binding site located in Sudlow’s site. Interactions with vicenin-2, particularly in its nonplanar conformation, lead to hydrophobic interactions involving the cinnamoyl system with a subpocket composed of Leu198, Lys199, Ser202, Phe211, Trp214, Ala215, Arg218, and Leu481, in addition to Pi-pi stacking between the phenyl ring and the side chain of Trp214, as well as a hydrogen bond with the beta-glucosyl units. This process results in a structure that is able to overcome oxidative stress and heat-induced damage based on the molecular forces highlighted in our work, which are predominately enthalpy-driven, as shown by the variation in thermodynamic parameters.

## Figures and Tables

**Figure 1 ijms-24-17222-f001:**
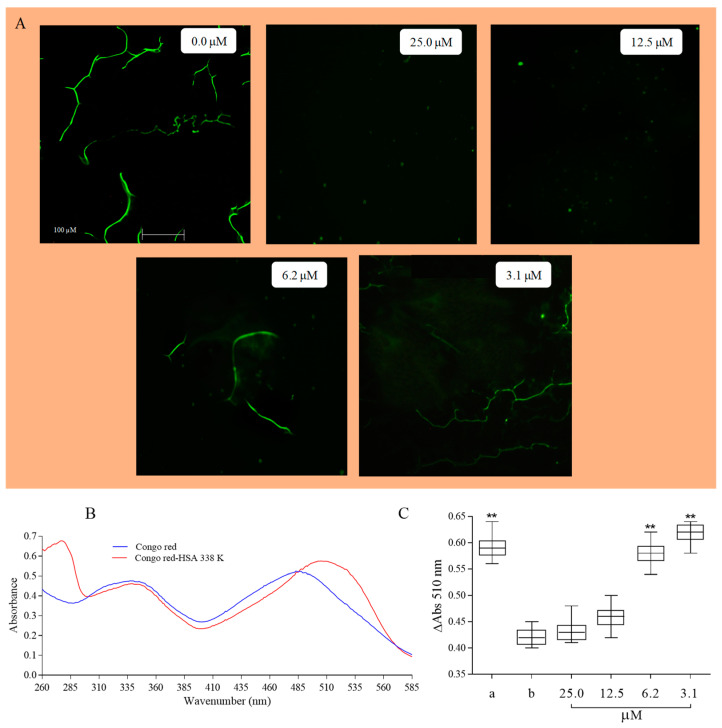
(**A**) Representative fluorescence microscopy images of HSA fibrils in the absence or presence (0.0–25.0 µM) of vicenin-2. (**B**) UV-visible spectra of Congo red alone or with HSA treated at high temperature. (**C**) Variation in the maximum absorbance of Congo red–HSA in the absence or presence of 0.0–25.0 µM of vicenin-2 after treatment to induce fibrillation. (a) HSA alone incubated at 338 K; (b) HSA alone incubated at 310 K. Asterisks (**) indicate a significant difference with respect to control (*p* < 0.05).

**Figure 2 ijms-24-17222-f002:**
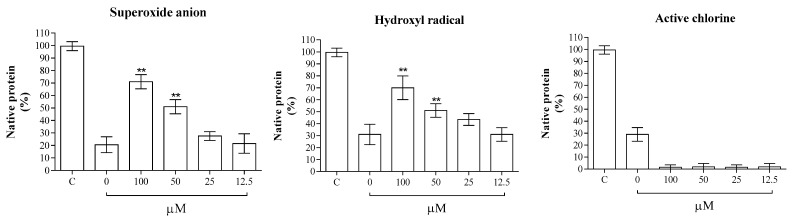
The impact of vicenin-2 on protein degradation induced by superoxide anion, hydroxyl radical, and active chlorine. HSA was subjected to electrophoretic separation on polyacrylamide gel electrophoresis (PAGE), with incubation in the absence or presence of different radical alone, or with the addition of 12.5–100.0 μM of the flavonoid. The samples were subjected a 40 min incubation at 310 K and were subsequently analyzed by 7.5% polyacrylamide-gel electrophoresisThe integrated density of each band is presented as a percentage of the untreated HSA sample in the experiments involving superoxide anion, hydroxyl radical, and active chlorine. The histograms depict the data as means ± S.D. (n = 3). Asterisks (**) indicate a significant difference with respect to control (*p* < 0.05).

**Figure 3 ijms-24-17222-f003:**
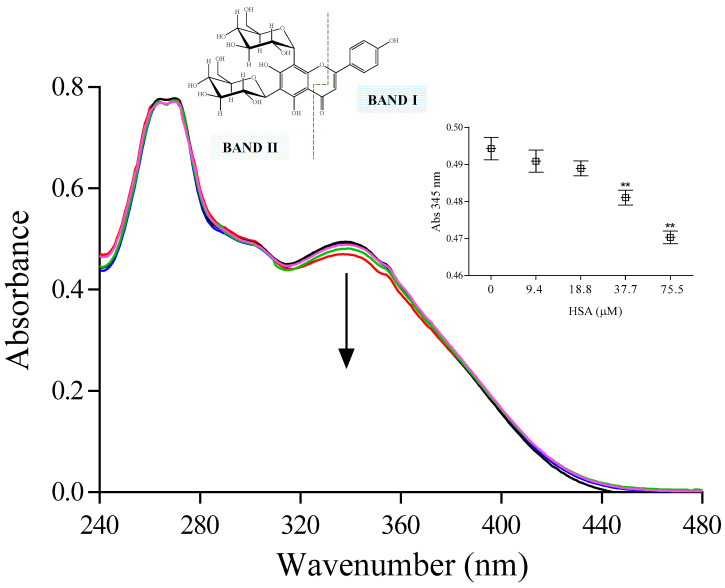
UV-visible absorption spectra of vicenin-2 (**—** 76.5 µM) in the absence or presence of increasing HSA concentrations (**—** 9.4 µM, **—** 18.8 µM, **—** 37.7 µM, **—** 75.5 µM). The inset shows the variation of absorbance obtained with three different experiments at the maximum absorbance of Band I. Asterisks (**) indicate a significant difference with respect to control (*p* < 0.05).

**Figure 4 ijms-24-17222-f004:**
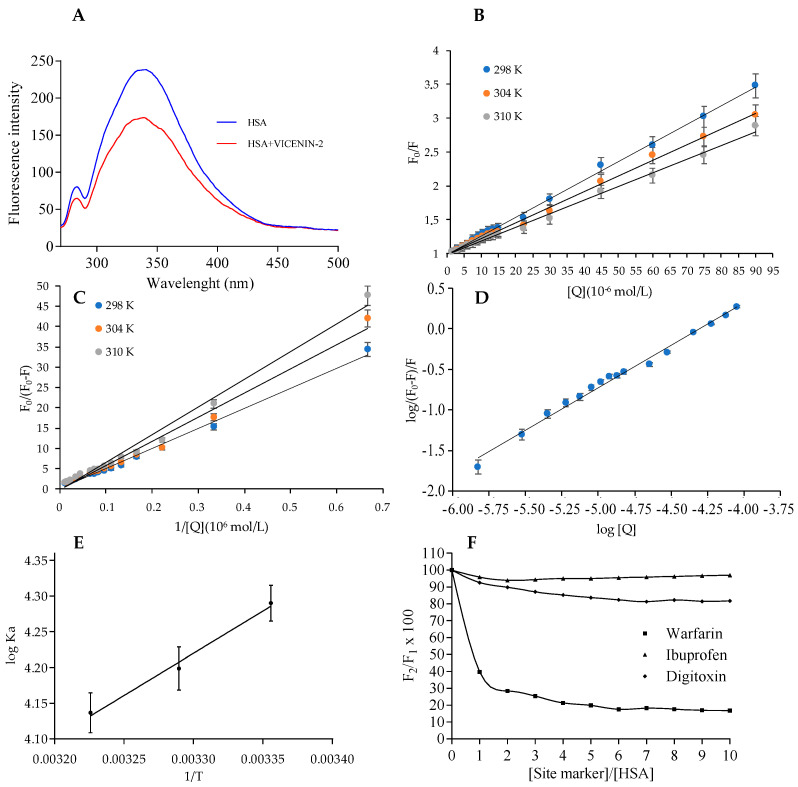
Fluorescence emission spectra of HSA (1.5 × 10^−5^ mol/L) in the absence or presence of the same molar concentration (1.5 × 10^−5^ mol/L) of vicenin-2 at 298 K and the maximum tested concentration of the flavonoid (**A**). Stern–Volmer (**B**) and modified Stern–Volmer (**C**) plots for the vicenin-2–HSA complex at three different temperatures. Analysis of binding equilibrium, thermodynamics, and acting forces. Plots of log(*F*0 − *F*)/*F* as a function of *log*[*Q*] for the binding of vicenin-2 to HSA at the temperature of 310 K (**D**), as well as van’t Hoff plot (**E**) and effect of site-specific markers on the fluorescence of HSA–vicenin-2 complex (**F**). Data are the results of three different experiments, expressed as mean ± SD.

**Figure 5 ijms-24-17222-f005:**
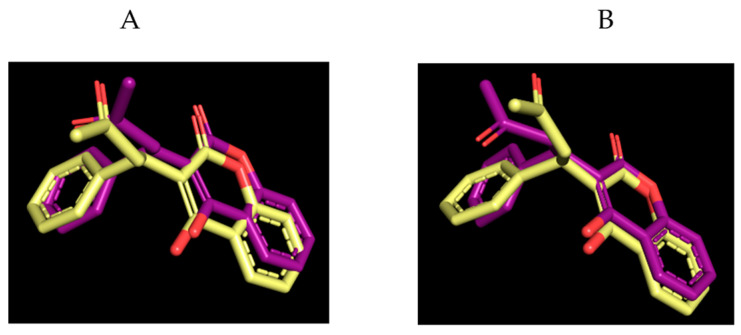
Superimposition of the crystalized warfarin (yellow sticks) to the rigid docking (**A**) and the induced fit docking (**B**) poses (purple sticks).

**Figure 6 ijms-24-17222-f006:**
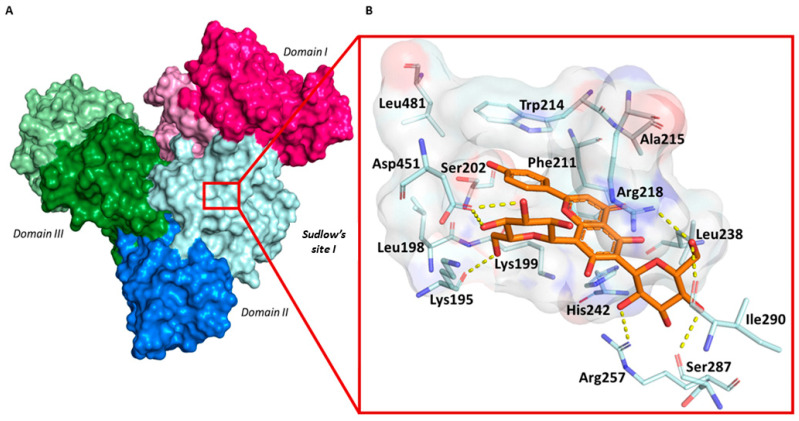
An overview of the HSA 3D structure and identification of Sudlow’s site I (**A**). Magnification focused on the binding site and the interactions of the vicenin-2 with the residues forming the pockets (**B**).

**Figure 7 ijms-24-17222-f007:**
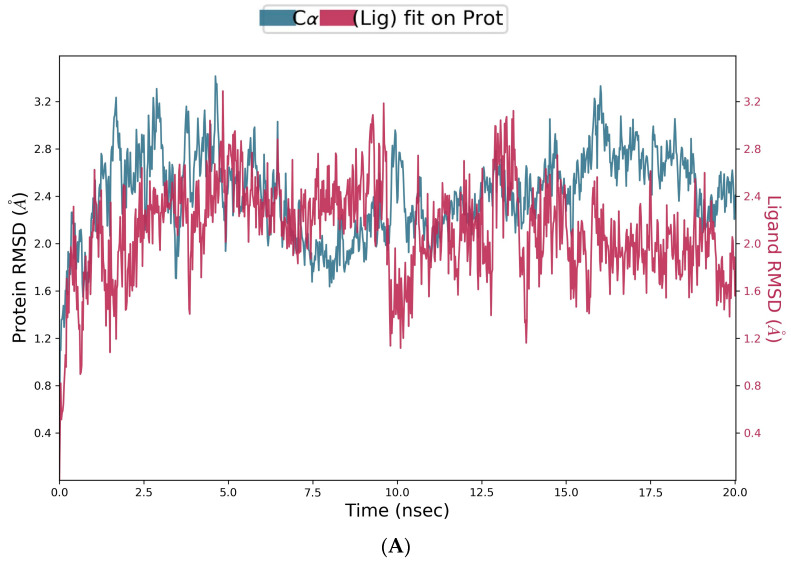
The plots of RMSD of the complex HSA–vicenin-2 (**A**) and the free HSA (**B**). On the x-axis, the simulation time is depicted. The left y-axis illustrates the HAS RMSD progression, while the y-axis on the right plots the RMSD evolution of vicenin-2 within the binding site during the simulation, indicating the degree of stability exhibited by vicenin-2.

**Figure 8 ijms-24-17222-f008:**
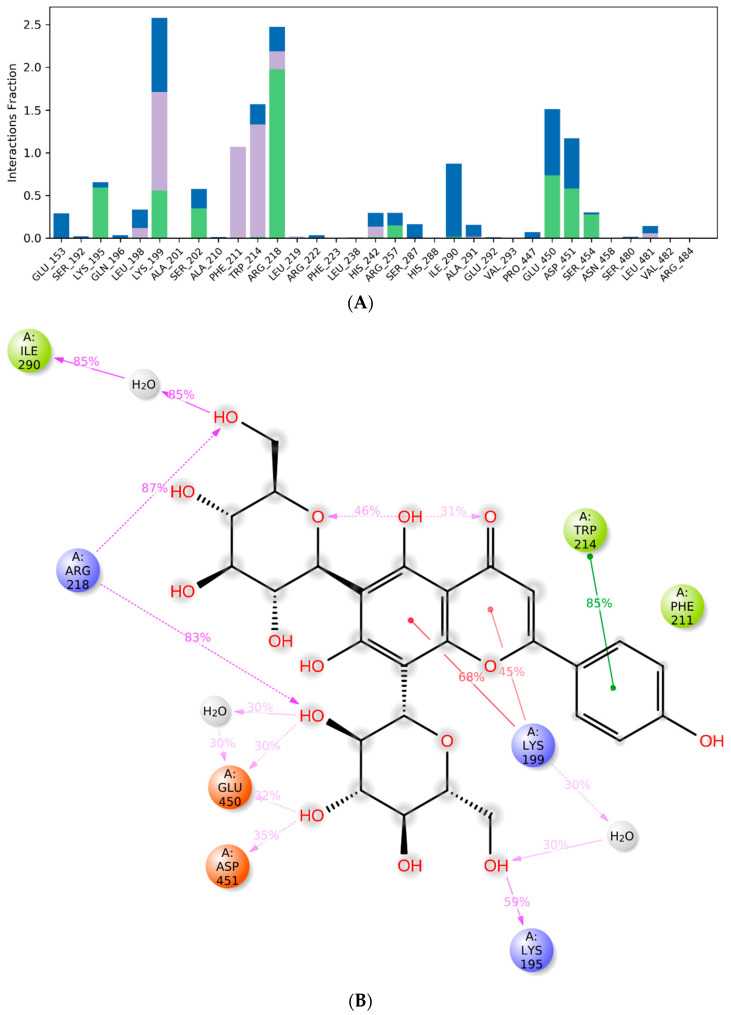
Molecular interaction in the complex. (**A**) Protein–ligand interactions are classified into four categories distinguishable by the color of the bars: hydrophobic interactions (purple), hydrogen bonds (green), ionic contacts (fuchsia), and water bridges (blue). (**B**) An in-depth examination of the contacts between ligand atoms and residues, considering interactions that occur more than 40% of the time.

**Figure 9 ijms-24-17222-f009:**
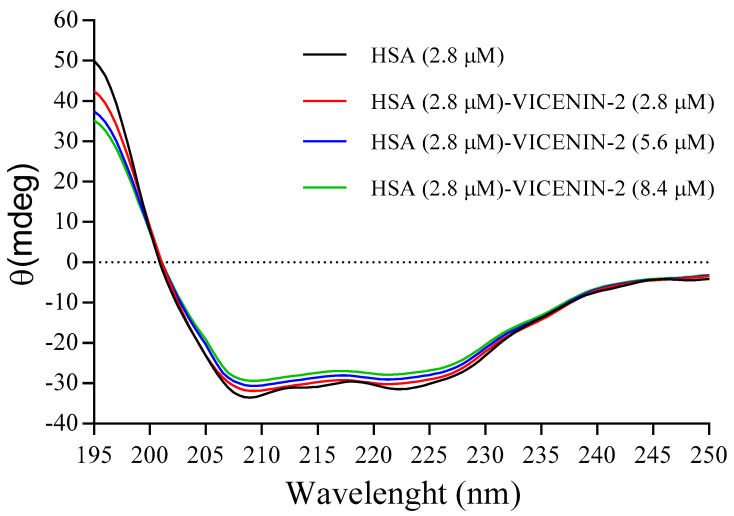
CD spectra of free HSA and the vicenin-2/HSA complex at T = 298 K in phosphate buffer of pH 7.4, 20 mM.

**Table 1 ijms-24-17222-t001:** Constants for the interaction of vicenin-2 with HSA at different temperatures and Δ*G*.

*T* (K)	*K_SV_* (M^−1^)	*K_a_* (M^−1^)	Δ*G* (kJmol^−1^)
298	2.75(±0.10) × 10^4^	1.95(±0.12) × 10^4^	−11.49 ± 0.91
304	2.32(±0.09) × 10^4^	1.58(±0.08) × 10^4^	−11.52 ± 0.91
*310*	2.10(±0.07) × 10^4^	1.37(±0.10) × 10^4^	−11.55 ± 0.90

## Data Availability

Data are contained within the article.

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
