# Peer review of "Anti-Aggregative and Protective Effects of Vicenin-2 on Heat and Oxidative Stress-Induced Damage on Protein Structures"

_ijms, 2023, doi:10.3390/ijms242417222_

Round 1
Reviewer 1 Report
Comments and Suggestions for Authors
The current manuscript reports the investigation of the interactions between the flavonoid vicenin-2 and human serum albumin using a variety of spectroscopic approaches and computational methods in a complementary fashion. A concentration-dependent inhibition of HSA fiber formation and protection of this protein against the action of some oxidation agents by vicenin-2 are also uncovered here.
The aims of the work are clearly defined. The investigation of the interactions between HSA, the major protein in blood plasma, and chemicals modulating its self-association properties and its susceptibility to oxidation is considered relevant. The manuscript is well structured and the experiments presented in a logical fashion. I have, however, a number of concerns:
-In the UV-visible spectra shown in figure 3 only minor changes are observed upon HSA addition to the compound. Are the changes described in the text (lines 168-169) significant when average absorbance values (with errors) from replicates, with and without HSA, are compared?
-Was the 1:1 vicenin-2:HSA binding stoichiometry (line 182) obtained from the UV-vis titrations? If so, please, explain how this conclusion was reached.
-What are the concentrations of HSA in each of the spectrum shown in figure 3?
-In the materials and methods (lines 469-472) it is stated that the spectra of a solution with the same concentration of vicenin-2 was subtracted in the UV-vis spectra, but then the spectrum in the absence of HSA should be flat and the absorbance of HSA at 280 nm should be observed in the different spectra, and it should vary with the protein concentration. Was the spectrum of HSA at each concentration the one actually subtracted?
-The binding constants obtained from the fluorescence experiments (lines 239-240) correspond to dissociation constants in the order of 50-70 microM, depending on the temperature. However, the fluorescence titrations were conducted at much lower concentrations of vicenin-2 (up to 15 microM), meaning that the system is very far from saturation. In fact, according to the binding constants, only minimal binding would occur in the tested concentration range. It is puzzling how biding constants could be estimated under these conditions.
-In the fluorescence experiments, have the inner filter effects been taken into account? It seems from the spectra in figure 3 that vicenin-2 has significant absorbance.
-The number of replicates in the experiments shown in Figure 4 and the errors in the data and in the different constants retrieved from analysis of these experiments are missing.
-From the CD spectroscopy analysis, it is concluded that a 2.6% change in the alpha-helix content occurs upon binding of the compound. Is this change significant when averages from replicates and associated errors are considered? Besides, according to the binding constants obtained by fluorescence, at the concentrations used in the CD spectroscopy analysis (around 3 microM) the level of binding would be insignificant.
-A prior article: (Anal Bioanal Chem. 2008 May;391(2):625-32. Characterization of interactions between polyphenolic compounds and human serum proteins by capillary electrophoresis. Andréa Diniz, Laura Escuder-Gilabert, Norberto P Lopes, Rosa María Villanueva-Camañas, Salvador Sagrado, María José Medina-Hernández) showed that “vicenin-2 do not present affinity towards HSA (<10%)”. I strongly recommend the authors discussing this work.
-I suggest showing the inhibition of fiber formation by DIC, in addition to the fluorescence images shown here.
-In figures 1B and 3, please, label the X axis as “wavelength (nm)” instead of “wavenumber”.
-It is not clear to me what it is represented in figure 1C. The Y-axis is labelled as "delta abs 510", what was subtracted here? In the figure legend it is stated “Variation in the maximum of absorbance”, does this mean the position of the maximum (nm)? The values seem to correspond to the net absorbances of the spectra in figure 1B, under the two conditions shown.
-Is the protection from oxidation specific for HSA and dependent on the binding of the compound to the protein? Would the presence of vicenin-2 in the solution protect any protein from oxidation?
-What was the percentage of the DMSO accompanying vicenin-2 in the experiments? Was its effect on HSA in the different experiments verified?
-From the experiments in Figure 4F it is concluded that there is a competition between warfarin and vicenin-2 for the binding to site I (lines 289-290). I am not sure to understand how this conclusion was reached. If warfarin is still able to modify the fluorescence of HSA in the presence of vicenin-2, wouldn’t this mean that there is no competition instead? Conversely, if the other two compounds do not modify fluorescence in the presence of vicenin-2, perhaps vicenin-2 may be competing with them? I strongly recommend completing the description of these experiments by indicating clearly what F1 and F2 are, the wavelengths of excitation and emission and showing the same titrations in the absence of vicenin-2, and perhaps the actual spectra, maybe as supplementary material.
Minor issues:
-To improve readability, I suggest separating the information in the introduction section in different shorter paragraphs.
-The first lines (78-87) in section 2.1 seem a bit too general. I suggest better focusing on the aspects needed to understand the experiments performed and shifting the more general information to the introduction, for example the mention to crowding.
-Although most of the experimental conditions are stated in the materials and methods section, I think that including some of these data (for example the concentrations) in the figure legends would help the reader.
-In figure 1c, the actual concentrations could be indicated in the X axis, instead of the letter code.
-What is meant by “bimolecular complex” in line 127?
-The argumentation in lines 133-137 is difficult to follow. I suggest splitting the single sentence in two or more.
-Please, remove the “and” in line 145.
- Line 147: What is “PAGE inducing damage due to superoxide anion”?
-The 2.2 title may be improved, it is not clear what the “process” is.
-The text in lines 237-250 is difficult to read.
-The equations in the section where the fluorescence experiments are described could be shifted to materials and methods.
-Figure 2A: please, specify the concentration of vicenin-2 in each lane.
-In line 295, should it be Figure 4F instead of Figure 4E?
Comments on the Quality of English Language
English seems generally fine.
Author Response
The current manuscript reports the investigation of the interactions between the flavonoid vicenin-2 and human serum albumin using a variety of spectroscopic approaches and computational methods in a complementary fashion. A concentration-dependent inhibition of HSA fiber formation and protection of this protein against the action of some oxidation agents by vicenin-2 are also uncovered here.
The aims of the work are clearly defined. The investigation of the interactions between HSA, the major protein in blood plasma, and chemicals modulating its self-association properties and its susceptibility to oxidation is considered relevant. The manuscript is well structured and the experiments presented in a logical fashion.
Thank you for the evaluation and for the very useful following comments.
I have, however, a number of concerns:
-In the UV-visible spectra shown in figure 3 only minor changes are observed upon HSA addition to the compound. Are the changes described in the text (lines 168-169) significant when average absorbance values (with errors) from replicates, with and without HSA, are compared?
According to reviewer suggestion we have added a graph that show the average values±SD, obtained from three different experiment performed in the same experimental condition.
-Was the 1:1 vicenin-2:HSA binding stoichiometry (line 182) obtained from the UV-vis titrations? If so, please, explain how this conclusion was reached.
Yes, the first evidence has been obtained by UV-Vis spectroscopy. We have added the following sentence to the text to explain this data: “ The titration of vicenin-2 with increasing HSA concentrations superior than 76.5 µM (data not shown) did not shown any significant change in the absorption band, suggesting that all the vicenin-2 molecules are already involved in the interaction with HSA. These results let us to suppose that the interaction between the flavonoid and HSA can be described with 1:1 stoichiometry.”
-What are the concentrations of HSA in each of the spectrum shown in figure 3?
According to reviewer suggestion we have added the concentration of HSA.
-In the materials and methods (lines 469-472) it is stated that the spectra of a solution with the same concentration of vicenin-2 was subtracted in the UV-vis spectra, but then the spectrum in the absence of HSA should be flat and the absorbance of HSA at 280 nm should be observed in the different spectra, and it should vary with the protein concentration. Was the spectrum of HSA at each concentration the one actually subtracted?
Thank you for the suggestion there was a mistake in the draft of the work. The subtracted spectrum is the one of HSA as you correctly suggested. We have modified, according to your suggestion, the text in the corresponding section of the work.
-The binding constants obtained from the fluorescence experiments (lines 239-240) correspond to dissociation constants in the order of 50-70 microM, depending on the temperature. However, the fluorescence titrations were conducted at much lower concentrations of vicenin-2 (up to 15 microM), meaning that the system is very far from saturation. In fact, according to the binding constants, only minimal binding would occur in the tested concentration range. It is puzzling how biding constants could be estimated under these conditions.
According to reviewer suggestion we have added the data obtained from fluorescence titrations up to 90 microM.
-In the fluorescence experiments, have the inner filter effects been taken into account? It seems from the spectra in figure 3 that vicenin-2 has significant absorbance.
Thank you for the useful comment. Yes, we have taking into accounts the inner filter effects, but we don’t have reported it in the materials and methods section. Now we have added the following sentence:
“All the fluorescence intensities were corrected for the inner filter effect utilizing the following formula:
Fcorr=Fobsx10(Aex-Aem)/2
where, Fcorr, Fobs, Aex and Aem are the corrected and observed fluorescence intensity, and the absorbance at excitation and emission wavelength, respectively [41].”
-The number of replicates in the experiments shown in Figure 4 and the errors in the data and in the different constants retrieved from analysis of these experiments are missing.
We have modified the graph and the constants according to the reviewer suggestion.
-From the CD spectroscopy analysis, it is concluded that a 2.6% change in the alpha-helix content occurs upon binding of the compound. Is this change significant when averages from replicates and associated errors are considered? Besides, according to the binding constants obtained by fluorescence, at the concentrations used in the CD spectroscopy analysis (around 3 microM) the level of binding would be insignificant.
We modify the value as mean±SD by performing three different experiment. We want to analyze the interaction between vicenin-2 and HSA that happens when they react at the same concentration, for this reason we utilize this low concentration, because this concentration of HSA give a good CD spectrum.
-A prior article: (Anal Bioanal Chem. 2008 May;391(2):625-32. Characterization of interactions between polyphenolic compounds and human serum proteins by capillary electrophoresis. Andréa Diniz, Laura Escuder-Gilabert, Norberto P Lopes, Rosa María Villanueva-Camañas, Salvador Sagrado, María José Medina-Hernández) showed that “vicenin-2 do not present affinity towards HSA (<10%)”. I strongly recommend the authors discussing this work.
According to reviewer suggestion we have added the following discussion to the text:
“These data integrate the data of Diniz et al., that reported a low percentage of binding to HSA of vicenin-2 detected by capillary electrophoresis and are not able to estimate the affinity constants and the numbers of binding primary sites because the obtained experimental data did not fit the equation utilized for the determination due mainly to the low hydrophobicity of vicenin-2.”
-I suggest showing the inhibition of fiber formation by DIC, in addition to the fluorescence images shown here.
Thank you for the suggestion that we will utilize for the preparation of a new work, but in this work we haven’t the possibility to perform this analysis.
-In figures 1B and 3, please, label the X axis as “wavelength (nm)” instead of “wavenumber”.
Done
-It is not clear to me what it is represented in figure 1C. The Y-axis is labelled as "delta abs 510", what was subtracted here? In the figure legend it is stated “Variation in the maximum of absorbance”, does this mean the position of the maximum (nm)? The values seem to correspond to the net absorbances of the spectra in figure 1B, under the two conditions shown.
Yes, it is the maximum (nm) and correspond to the net absorbances of the spectra in figure 1B, under the two conditions shown, with the addition of the effects of different vicenin-2 concentrations.
-Is the protection from oxidation specific for HSA and dependent on the binding of the compound to the protein? Would the presence of vicenin-2 in the solution protect any protein from oxidation?
In our experiment the protection has been analysed following the binding, and it is a general process in which vicenin-2 binding to a protein. We utilize HSA because it is one of the most abundant protein in the organism and a model protein well studied and characterized. Vicenin-2 is able to have an antioxidant activity in solution, so, probably it can protect protein from oxidation.
-What was the percentage of the DMSO accompanying vicenin-2 in the experiments? Was its effect on HSA in the different experiments verified?
Yes, in all the performed experiment we have tested the effects of solvent to exclude its involvement in the analysed process and we have maintained its amounts at 0,5% or below.
-From the experiments in Figure 4F it is concluded that there is a competition between warfarin and vicenin-2 for the binding to site I (lines 289-290). I am not sure to understand how this conclusion was reached. If warfarin is still able to modify the fluorescence of HSA in the presence of vicenin-2, wouldn’t this mean that there is no competition instead? Conversely, if the other two compounds do not modify fluorescence in the presence of vicenin-2, perhaps vicenin-2 may be competing with them? I strongly recommend completing the description of these experiments by indicating clearly what F1 and F2 are, the wavelengths of excitation and emission and showing the same titrations in the absence of vicenin-2, and perhaps the actual spectra, maybe as supplementary material.
Thank you for the suggestion we have clearly define F1 and F2, the wavelengths of excitation and emission in the corresponding material and methods section. We have utilized a standard method of displacement (or competition) that is possible to find in a lot of works in literature and follows the operative indications, we haven’t analyze the theoretical basis of the process in our work because we want just to find the possible binding site, however in literature is possible to find the titration of HSA with the three well know site marker. We think is better to maintain the actual data obtained from the specta in the main body of the work, rather than to report them as supplementary material.
Minor issues:
-To improve readability, I suggest separating the information in the introduction section in different shorter paragraphs.
According to reviewer suggestion we have modify the introduction.
-The first lines (78-87) in section 2.1 seem a bit too general. I suggest better focusing on the aspects needed to understand the experiments performed and shifting the more general information to the introduction, for example the mention to crowding.
According to reviewer suggestion we have modify the starting part of the section.
-Although most of the experimental conditions are stated in the materials and methods section, I think that including some of these data (for example the concentrations) in the figure legends would help the reader.
According to reviewer suggestion we have modify the legend.
-In figure 1c, the actual concentrations could be indicated in the X axis, instead of the letter code.
According to reviewer suggestion we have modify theX axis of figure 1c indicating the concentrations.
-What is meant by “bimolecular complex” in line 127?
Thank for the suggestion we have change “biomolecular complex” with “HSA-vicenin-2 complex”
-The argumentation in lines 133-137 is difficult to follow. I suggest splitting the single sentence in two or more.
Done.
-Please, remove the “and” in line 145.
Done.
- Line 147: What is “PAGE inducing damage due to superoxide anion”?
According to reviewer suggestion we have modify the sentence.
-The 2.2 title may be improved, it is not clear what the “process” is.
Done.
-The text in lines 237-250 is difficult to read.
According to reviewer suggestion we have modify the sentence.
-The equations in the section where the fluorescence experiments are described could be shifted to materials and methods.
We prefer to maintain them in the text, to make the text more reader-friendly.
-Figure 2A: please, specify the concentration of vicenin-2 in each lane.
Done.
-In line 295, should it be Figure 4F instead of Figure 4E?
Thank you for the suggestion we have modify accordingly.
Reviewer 2 Report
Comments and Suggestions for Authors
Dear Authors,
The manuscript “Anti-Aggregative and Protective Effects of Vicenin-2 on Heat and Oxidative Stress Induced Damage on Protein Structures.” was written by Giuseppe Tancredi Patanè and co-authored. The topic of flavonoid impacts in the biological processes is an interesting, and contemporary area of research. Many of these types of molecules, which are obtained from natural plant sources, have found application in pharmacy. The article has a good overall impression. It has an appropriate sectional structure for this type of research, which is presented in a concise, and clear manner. The introduction is quite general and could be expanded upon. The discussion of research results is justified and supported by experiments. However, the organization and some figures could be presented in better quality. I have included all my comments in detail in the list below, please respond to comments, which, in my opinion, will significantly improve the quality of the manuscript.
1. Please explain specifically in what biological processes dysfunctional albumin aggregation (HSA) may occur to clearly prove the importance of this research. The authors refer to:
· Ajmal, M.R. Protein Misfolding and Aggregation in Proteinopathies: Causes, Mechanism and Cellular Response. Diseases. 2023, 608 11(1), 30. 609
· Wen, J.H.; He, X.H.; Feng, Z.S.; Li, D.L.; Tang, J.X.; Liu, H.F. Cellular Protein Aggregates: Formation, Biological Effects, and 610 Ways of Elimination. Int. J. Mol. Sci. 2023, 24(10), 8593.
These articles broadly discuss the aggregation of synuclein, b-amyloid, and tau protein, which are associated with neurodegenerative diseases. It should be explained why the authors investigated the effect of vicenin-2 on the inhibition of albumin aggregation (why exactly HSA).
2. Were fluorescence spectroscopy measurements carried out without inner-filter effect correction? The use of this correction may affect the obtained values of the KSV constants, especially for measurements at different temperatures.
3. I recommend creating the table with the fluorescence spectroscopy data.
4. The conformational changes of HSA after vicenin-2 binding were monitored only by reducing the a-helix. The authors used the Jasco J-810 CD spectrometer which software allows to calculation of more types of secondary strain than a-helix. As can be seen in the CD spectrum, the shape is different after binding to vicenin-2, so I recommend conducting a detailed conformational analysis. Of course, you can use some other software or online application.
5. Sometimes author used to specify temperature ‘°C’ and sometimes ‘K’. please standardize the way of writing the temperature unit throughout the text.
6. The resolution of the figures should be higher to make the data easier to read. I also recommend reorganizing some of them. For example:
· Figure 1- part B and C should be separated, drawing one by one, and enlargement plots will make them more readable.
· Figure 2 – the higher resolution on part C is required.
· Figure 3 – please add the legend of different HAS concentrations.
· Figure 7 – higher resolution is required.
· Figure 8 – higher resolution is required, the arrows are unreadable.
7. Please read carefully the whole article with attention to editorial and typos bags. In my opinion, there are small and do not diminish work, but should be corrected, e.g.:
· line 184 – italic “Figure 3”, it should be the same font format in the whole paper;
· line 195 – too many commas and space;
· line 227 – upper index (104);
· line 254 – I recommend using a small letter in ‘log’ (see equation)?
· line 256 – lower index (F0);
· line 266 – ‘(eq 3):’ actually which equation does it refer to?
· line 274 – upper index (mol-1);
· line 328 – it school be ‘kcal/mol’ (small letter);
· line 380 – the molecular mass require unit;
· the equations in text should be numbered.
8. I am concerned about one more issue, in the Author Contributions section it is written that: "funding acquisition, D.B." but in the Funding part there is no specified funding. Maybe you just missed something vital issue to fill it up.
Comments on the Quality of English LanguageThe English is fine and scientific sound.
Author Response
The manuscript “Anti-Aggregative and Protective Effects of Vicenin-2 on Heat and Oxidative Stress Induced Damage on Protein Structures.” was written by Giuseppe Tancredi Patanè and co-authored. The topic of flavonoid impacts in the biological processes is an interesting, and contemporary area of research. Many of these types of molecules, which are obtained from natural plant sources, have found application in pharmacy. The article has a good overall impression. It has an appropriate sectional structure for this type of research, which is presented in a concise, and clear manner. The introduction is quite general and could be expanded upon. The discussion of research results is justified and supported by experiments. However, the organization and some figures could be presented in better quality.
I have included all my comments in detail in the list below, please respond to comments, which, in my opinion, will significantly improve the quality of the manuscript.
Thank you for the evaluation and for the very useful following comments.
- Please explain specifically in what biological processes dysfunctional albumin aggregation (HSA) may occur to clearly prove the importance of this research. The authors refer to:
- Ajmal, M.R. Protein Misfolding and Aggregation in Proteinopathies: Causes, Mechanism and Cellular Response. Diseases. 2023, 608 11(1), 30. 609
- Wen, J.H.; He, X.H.; Feng, Z.S.; Li, D.L.; Tang, J.X.; Liu, H.F. Cellular Protein Aggregates: Formation, Biological Effects, and 610 Ways of Elimination. Int. J. Mol. Sci. 2023, 24(10), 8593.
These articles broadly discuss the aggregation of synuclein, b-amyloid, and tau protein, which are associated with neurodegenerative diseases. It should be explained why the authors investigated the effect of vicenin-2 on the inhibition of albumin aggregation (why exactly HSA).
According to reviewer suggestion we have added the following discussion to the text:
“Human serum albumin has been chosen to study the process because its physiological importance (it is one of the most abundant protein, it plays fundamental biologic roles such as carrier protein of exogenous and endogenous compounds and blood pressure regulator) and it has a well-defined and complex tertiary structure without any propensity to fibrillation in its native state, while, in vitro, it easily aggregates under different conditions crating structures with features characteristic of disease-associated amyloidogenesis conditions.”
- Were fluorescence spectroscopy measurements carried out without inner-filter effect correction? The use of this correction may affect the obtained values of the KSVconstants, especially for measurements at different temperatures.
Thank you for the useful comment. Yes, we have taking into accounts the inner filter effects, but we don’t have reported it in the materials and methods section. Now we have added the following sentence:
“All the fluorescence intensities were corrected for the inner filter effect utilizing the following formula:
Fcorr=Fobsx10(Aex-Aem)/2
where, Fcorr, Fobs, Aex and Aem are the corrected and observed fluorescence intensity, and the absorbance at excitation and emission wavelength, respectively [41].”
- I recommend creating the table with the fluorescence spectroscopy data.
According to reviewer suggestion we have added a table
- The conformational changes of HSA after vicenin-2 binding were monitored only by reducing the a-helix. The authors used the Jasco J-810 CD spectrometer which software allows to calculation of more types of secondary strain than a-helix. As can be seen in the CD spectrum, the shape is different after binding to vicenin-2, so I recommend conducting a detailed conformational analysis. Of course, you can use some other software or online application.
Thank you for your comments. Initially we have had your same opinion but when we performed this analysis with the application “Bestsel” we have obtained results very similar with the reported data. For this reason we have reported this last analysis in the work, taking into accounts that HSA is a protein belonging to the class of “all alpha proteins” based on Structural Classification of Proteins (SCOP) and they are in agreement with the crystallographic data of the protein.
- Sometimes author used to specify temperature ‘°C’ and sometimes ‘K’. please standardize the way of writing the temperature unit throughout the text.
According to reviewer suggestion we have standardize.
- The resolution of the figures should be higher to make the data easier to read. I also recommend reorganizing some of them. For example:
- Figure 1- part B and C should be separated, drawing one by one, and enlargement plots will make them more readable.
According to reviewer suggestion we have re-organize the figure and enlarged all the plots.
- Figure 2 – the higher resolution on part C is required.
Done
- Figure 3 – please add the legend of different HAS concentrations.
Done.
- Figure 7 – higher resolution is required.9
Done.
- Figure 8 – higher resolution is required, the arrows are unreadable.
Done.
- Please read carefully the whole article with attention to editorial and typos bags. In my opinion, there are small and do not diminish work, but should be corrected, e.g.:
- line 184 – italic “Figure 3”, it should be the same font format in the whole paper;
Done.
- line 195 – too many commas and space;
Done.
- line 227 – upper index (104);
Done.
- line 254 – I recommend using a small letter in ‘log’ (see equation)?
Done.
- line 256 – lower index (F0);
Done.
- line 266 – ‘(eq 3):’ actually which equation does it refer to?
Deleted.
- line 274 – upper index (mol-1);
Done.
- line 328 – it school be ‘kcal/mol’ (small letter);
Done.
- line 380 – the molecular mass require unit;
Done.
- the equations in text should be numbered.
Done.
- I am concerned about one more issue, in the Author Contributions section it is written that: "funding acquisition, D.B." but in the Funding part there is no specified funding. Maybe you just missed something vital issue to fill it up.
It is with reference to the funding acquisition to pay the APC, that has been acquired
Round 2
Reviewer 1 Report
Comments and Suggestions for Authors
The authors have addressed most of my concerns in the revised version of the manuscript. I really appreciate the changes made. I still have a couple of comments, though. Regarding the CD spectroscopy analysis, even if the concentration of HSA used is convenient to obtain a CD spectrum, the concentrations of both vicenin-2 and of HSA seem too low to achieve significant binding, according to the binding constants determined in this work. I strongly recommend repeating the experiment at higher concentrations of at least one of the interacting species to populate the protein-ligand complexes. Alternatively, the authors can reconsider the significance of this experiment, as it could underestimate changes occurring upon substantial binding of the ligand to HSA. Besides, I suggest adding the spectrum at 90 microM vicenin-2 in Figure 4A.
Comments on the Quality of English LanguageOnly minor edits required
Author Response
The authors have addressed most of my concerns in the revised version of the manuscript. I really appreciate the changes made.
Thank you dear reviewer, we also really appreciate your suggestions and we believe that they have contributed to increase the quality of the work.
I still have a couple of comments, though. Regarding the CD spectroscopy analysis, even if the concentration of HSA used is convenient to obtain a CD spectrum, the concentrations of both vicenin-2 and of HSA seem too low to achieve significant binding, according to the binding constants determined in this work. I strongly recommend repeating the experiment at higher concentrations of at least one of the interacting species to populate the protein-ligand complexes. Alternatively, the authors can reconsider the significance of this experiment, as it could underestimate changes occurring upon substantial binding of the ligand to HSA.
According to the reviewer suggestion we have carried out the experiments with also the double and the triple concentration of vicenin-2 with respect to the one of HSA and added the graph and the corresponding comments to the main text.
Besides, I suggest adding the spectrum at 90 microM vicenin-2 in Figure 4A.
According to the reviewer suggestion we have added the required spectrum at 90 microM vicenin-2 in Figure 4A.